

# Electron backscatter diffraction analysis unveils foraminiferal calcite microstructure and processes of diagenetic alteration

Frances A. Procter[1], Sandra Piazolo[1], Eleanor H. John[2], Richard Walshaw[1], Paul N. Pearson[2], Caroline H. Lear[2], Tracy Aze[1]

[1]School of Earth and Environment, University of Leeds, Leeds, LS2 9JT, UK
[2]School of Earth and Environmental Sciences, Cardiff University, Cardiff, CF10 3AT, UK

*Correspondence to*: Frances A. Procter (gy14fap@leeds.ac.uk)

**Abstract**

Electron backscatter diffraction (EBSD) analysis enables a unique perspective of the internal microstructure of foraminiferal calcite. Specifically, EBSD provides crystallographic data from within the test, highlighting the highly organised "mesocrystal" structure of crystallographically aligned domains throughout the test, formed by sequential deposits of microgranular calcite. We compared EBSD maps across the test walls of both poorly- and well-preserved specimens of the planktonic foraminifera species *Globigerinoides ruber* and *Morozovella crater*. The EBSD maps, paired with information about intra-test distributions of Mg/Ca ratios, allowed us to examine the effects of different diagenetic processes on the foraminifera test. In poorly-preserved specimens EBSD data shows extensive reorganisation of the biogenic crystal microstructure, indicating differing phases of dissolution, re-precipitation and overgrowth. The specimens with the greatest degree of microstructural reorganisation also show an absence of higher concentration magnesium bands, which are typical features of well-preserved specimens. These findings provide important insights into the extent of post-depositional changes both in microstructure and geochemical signals that must be considered when utilising foraminifera to generate proxy archive data.

## 1 Introduction

The fossilised shells (tests) of foraminifera provide some of the most widely used and invaluable climate proxy data available to the Earth and environmental sciences (e.g. Zachos et al., 2001; Zachos, Dickens and Zeebe, 2008; Westerhold et al., 2020), primarily due to their geographic extent and the unparalleled temporal resolution of their fossil record. Geochemical analysis of stable isotope ratios such as carbon and oxygen, and trace element concentrations, such as the ratio of magnesium to calcium (Mg/Ca) in foraminiferal calcite can provide reliable estimates of past global climates (e.g. Erez and Luz, 1983; Nürnberg, Bijma, and Hemleben, 1996; Lea, Pak and Spero, 2000; Leng, 2006; Ganssen et al., 2011; Pearson, 2012). Despite the widespread application of planktic foraminifera in palaeoceanography and palaeoclimate reconstructions, the exact processes involved in the biomineralisation of their calcite tests are still uncertain (de Nooijer et al., 2014; Fehrenbacher et al., 2017; Arns et al., 2022; Lastam et al., 2023[a,b]). Restricted understanding of foraminiferal biomineralisation hinders our use of palaeo-



proxies; for example, Mg heterogeneity in foraminifera in the form of high and low Mg bands is believed to be biologically controlled, thus inducing uncertainty to the Mg/Ca temperature proxy (e.g. Eggins, Sadekov and De Deckker, 2004; Spero et al., 2015; John et al., 2023). Also, post-depositional processes can modify test microstructure, and this may be accompanied

by changes in the geochemistry of the test (Pearson et al., 2001; Sexton et al., 2006; Kozdon et al., 2013; Edgar et al., 2015; Staudigel et al., 2022; John et al., 2023). Not all fossil specimens are equally affected by diagenetic modifications, arguably due to differing depositional histories and environments (e.g. Parker and Berger, 1971) and original test structure and composition (e.g. Edgar, Pälike and Wilson, 2013). Previous work has demonstrated foraminifera that exhibit good ("glassy") preservation show limited signs of test wall degradation and presumably original geochemistry signals (e.g., Norris and Wilson,

1998; Pearson et al., 2001; Wendler et al., 2013). Conversely, poorly preserved ("frosty") specimens show significant test wall alteration and potentially significant impacts on geochemical proxy reliability (Pearson et al., 2001, 2007; Sexton et al., 2006; Pearson and Burgess, 2008). The exact relationship between microstructural alteration and geochemical change is still poorly understood. For example, Mg banding is preserved in some visibly altered foraminifera, suggesting that alteration may not necessarily preclude the use of some foraminifera for Mg thermometry (e.g. Staudigel et al., 2022). From SEM images alone,

test microstructure may look significantly altered, and thus these specimens would be excluded from use in geochemical proxy generation. Contrarily, some experimental studies suggest that even 'pristine' looking foraminifera tests may have undergone minor amounts of diagenetic ion exchange (Cisneros-Lazaro et al., 2022).

Electron backscatter diffraction analysis (EBSD) provides spatially defined, quantitative crystallographic orientation data that

can be used to examine primary features of foraminifera test walls and therefore the degree of microstructural change associated with diagenesis. By comparing such characterisations with intra-test geochemical variations, we can obtain insights into the alteration processes at individual sites, and better understand the relationship between microstructural change and geochemical alteration. EBSD analysis quantifies the full crystallographic orientation of a crystalline sample i.e. all crystallographic axes are determined (e.g. Prior et al., 1999). In recent years the resolution of the technique has improved,

which has resulted in its application to investigate both benthic and planktonic foraminifera microstructures (e.g. Pabich, Vollmen and Gussone, 2020; Yin et al., 2021; Lastam et al., 2023[a,b]).

Foraminifera precipitate their calcite tests as biogenic microgranules that are typically submicron in size, which coalesce to form larger structures (Hemleben et al. 1989; Schiebel and Hemleben, 2017). These microgranules are not currently

individually resolvable via EBSD mapping; instead this technique maps "grains", which represent areas of crystallographic alignment. In this contribution we define a grain as a domain of similar crystallographic orientation surrounded by a grain boundary with >10° orientation change between adjacent grains. These grains are typically synonymous with previously identified "mesocrystals" (Yin et al., 2021; Arns et al., 2022); and are composed of numerous small crystals of similar size and shape (in this case, the sub-micron microgranules of biogenic foraminiferal calcite). Some of the key findings of previous

EBSD investigations into biomineralisation processes include: 1) the "mesocrystal structure" of units of similarly oriented



nanometer scale microgranules; 2) the preferred alignment of the calcite *c*-axis perpendicular to the test wall, which is thought to be induced by biopolymers of the primary organic sheet (POS); and 3) the presence of 60° twinning in test calcite (e.g. Pabich, Vollmen and Gussone, 2020; Yin et al., 2021; Arns et al., 2022; Lastam et al., 2023[a,b]).

So far, this quantitative microstructural work has been limited to a small number of foraminifera species, even though it is well documented that the test wall characteristics are highly variable across different species of planktonic foraminifera. To date, there has been no work that has taken advantage of EBSD as a tool to investigate the impact of diagenetic alteration on planktonic foraminifera test microstructure. In this study, we use EBSD to investigate the crystallographic microstructural characteristics of two species of planktonic foraminifera with different wall textures and different post-depositional histories.

Specifically, we present data from *Globigerinoides ruber*, an extant species with a cancellate wall texture comprising an irregular honeycomb appearance on the surface where interpore ridges are present between pores on the surface of the test (Hembleben and Olsson, 2006). We also examine *Morozovella crater*, an extinct species with a muricate wall texture, characterised by surface projections called "muricae", formed by upward deflection and mounding of successive layers of calcite in the test wall (Pearson et al., 2022). Both species are frequently used in the generation of geochemical paleoclimate

proxy records (e.g. Lea, Pak and Spero, 2000; Hines et al., 2017). We analysed specimens displaying both glassy and frosty preservation, as indicated by examination under SEM.

Our EBSD measurements are coupled with maps of Mg/Ca ratios generated using electron microprobe analysis (EMPA), from the same specimens to examine the relationship between microstructural alterations and changes in geochemistry. We focus

on Mg/Ca ratios for the geochemical analysis as this proxy is commonly used in palaeoclimate reconstructions and is thought to be more robust to diagenetic changes than the oxygen isotope proxy (Kozdon et al., 2013; Staudigel et al., 2022). We aim to utilise these analytical techniques to illuminate the mechanisms involved in diagenetic alteration, and to evaluate the extent to which post-depositional history and foraminiferal wall texture may influence these processes.

## 2 Methods and materials

**2.1 Sample material and preparation**

We selected representative "glassy" and "frosty" specimens of each of the species *G. ruber* and *M. crater*. Both are taxa that lived in the surface mixed-layer of the open oceans and hosted algal photosymbionts (e.g. Aze et al., 2011). *G. ruber* (0–23 Ma) represents a Neogene to recent species with a cancellate wall texture and *M. crater* (43–53 Ma) is a Paleogene species with a muricate wall texture. Foraminifera were recovered from core material known to contain foraminifera specimens with

differing degrees of preservation (Fig. 1; Table 1). Representative SEM images from each sample are presented in Fig 2. Sediments were wet sieved over a 63 μm mesh, with residues then dried in an oven overnight at ~40ºC. Individual specimens of differing preservation quality were identified by visual inspection under a Zeiss Axio Zoom V16 light microscope. The



selected specimens were then ultrasonicated in DI water (and methanol for *M. crater* specimens) to remove any sediment residues. The preservation state of each sample was examined by scanning electron microscope (SEM) imaging of test fragments on an FEI Quanta 650 FEG ESEM (School of Earth and Environment, University of Leeds) and a Zeiss Sigma HD Field Emission Gun Analytical SEM (School of Earth and Environmental Sciences, Cardiff University). Four foraminifera specimens (both species, glassy and frosty) were then embedded in Epo Thin 2 epoxy resin using a 25 mm diameter mould. The resin block was ground gently using 1200 grit silicon carbide grinding disks, until the samples were exposed and the block face flat. The mounted sample was then polished using 6 µm, 3 µm, 1 µm and 0.25 µm diamond pastes, before a final polish for 8 minutes with 0.05 µm grade colloidal silica, creating a smooth surface at the equatorial plane of the specimens. Samples were carbon coated before SEM analysis and either carbon or silver coated for EMPA analyses, depending on the laboratory.

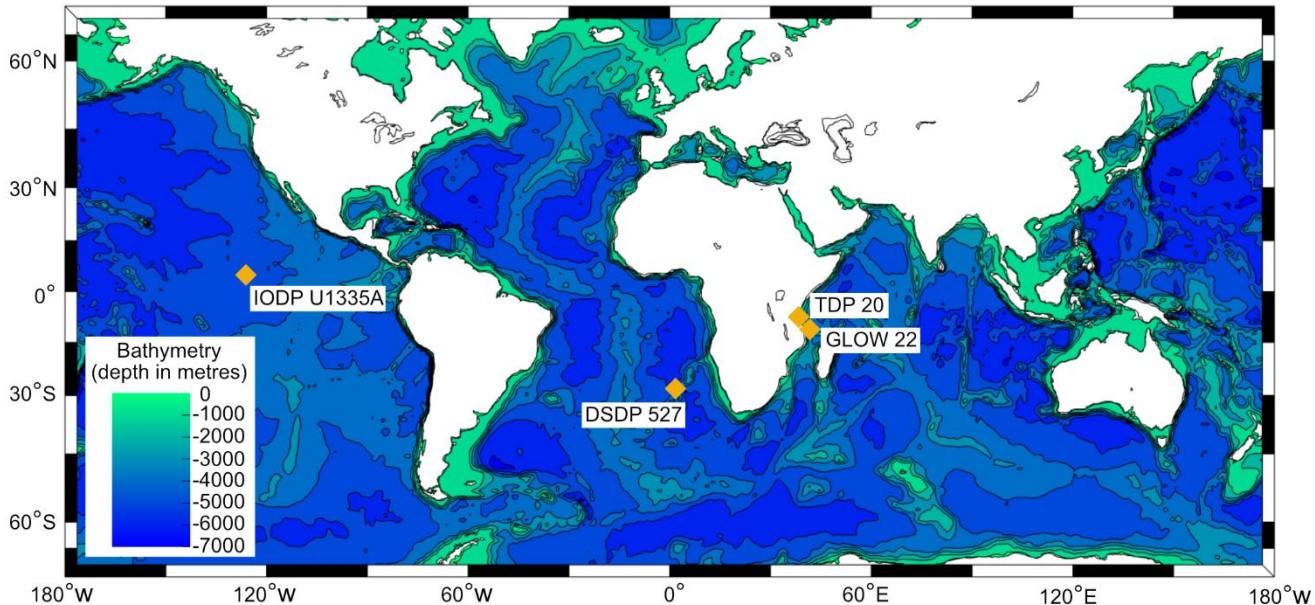

**Figure 1. Map showing the locality of sediment cores that provided specimens for analysis**

**2.2 Quantitative orientation and microstructural analysis using electron backscatter diffraction (EBSD) analysis**

EBSD was conducted on the FEI Quanta 650 FEG ESEM instrument, with the Oxford Instruments Symmetry detector (School of Earth and Environment, University of Leeds). The surface of the samples was inclined 70° to horizontal for analysis. Analyses were conducted at a voltage of 20 kV and a beam current of 8 nA. Comparable areas of the penultimate chamber of each specimen were indexed on a regular grid with a step size of 200 nm. Grains (crystallographically aligned domains) were mapped delimited by a defined grain boundary misorientation of >10° (where the difference in orientation between a grain and a neighbour grain, is greater than ten degrees). This paper focuses on these crystallographically aligned domains, or "mesocrystals". The ability to resolve intricate substructure was dictated by a precision of 0.3º in angular orientation of the



EBSD analysis itself, and a step size of 200 nm. If individual calcite structures were present within the shell, but were smaller than 200 nm, or differed in orientation from the neighbouring crystal by less than 0.3º, they would not have been resolved. Consequently, our study cannot inform on the presence and characteristics of such potential structures. Forescatter images

were also taken using the forescatter detectors mounted on the EBSD detector.

Table 1: Specimens analysed

| Species | Preservation | Wall Type | Host Sediment | Core Sample | ~ Age | Latitude/ Longitude | Site Reports |
|---|---|---|---|---|---|---|---|
| *G. ruber* | Glassy | Cancellate | Clay-rich | GLOW 22 Box Top | 4.39-5.49 Ma | 9°50'52.8"S 40°28'44.4"E | Kroon et al., 2009 |
| *G. ruber* | Frosty | Cancellate | Nanofossil pelagic ooze | IODP U1335A 13H 2 0-5 cm | 14 Ma | 5°18'44.1"N 126°17'00.1"W | Pälike et al., 2010 |
| *M. crater* | Glassy | Muricate | Clay-rich | TDP 20 32-1 45-51 cm | 47 Ma | 8°55'15.0"S 39°30'15.0"E | Nicholas et al., 2006 John et al., 2023 |
| *M. crater* | Frosty | Muricate | Nanofossil pelagic ooze | DSDP 527 18R 4 76-80 cm | 46-49 Ma | 28°02'29.4"S 1°45'48.0"E | Moore et al., 1984 |

**2.3 Electron microprobe analysis (EMPA)**

Elemental maps of the foraminifera tests were made for magnesium and calcium, using wavelength dispersive spectroscopy (WDS) to enable interpretation of chemical signal preservation. After EBSD analyses, in preparation for EMPA analyses, the sample blocks containing *G. ruber* were re-polished. Quantitative EMPA maps for *G. ruber* specimens were made at the University of Leeds on a JEOL JXA8230 Electron Probe Microanalyser, run at 15 kV and 20 nA with a focused beam. The dwell time was 3-4 seconds on peak, with a pixel size of 0.9 µm. Intensity measurements for the Mg Kα x-ray were collected

simultaneously across three WDS spectrometers, with the counts from all three being aggregated to improve counting statistics. All WDS maps were background corrected and quantitative maps were produced using the software package Calcimage (Probe





Software Inc, Eugene, US). Detection limits for Mg were 0.016 element wt% for an average of 4 pixels (or 0.031 element wt% for a single pixel). Analytical uncertainty expressed as S.E. ranged 7-13% for an average of 4 pixels across the displayed range in Mg concentration. EMPA maps for *M. crater* were made at the University of Bristol (also published in John et al.,

2023) using quantitative X-ray mapping on a JEOL 8530F field-emission electron microprobe, run at 15 kV and 80 nA using a focused beam, with a pixel size of 0.9 μm and a 500 ms dwell time. The detection limit for Mg for an average of 4 pixels was 0.015 wt% (or 0.031 wt% for a single pixel) to 3 s.d. Analytical uncertainty for an average of 4 pixels at 4 mmol/mol (close to the average map value) was ~9.5% S.E; while 11 pixels need to be integrated at lower Mg/Ca ratios of around 2 mmol/mol to achieve an error of 10%. Maps of *M. crater* were quantified in the Probe for EPMA software (Probe Software

Inc, Eugene, US). For all maps, data processing masked pixels with less than 35 wt% Ca, to remove pixels of non-pure calcium carbonate and to remove edge effects. ImageJ software was then used to present the maps in mmol/mol.

## 2.4 Data treatment

Electron backscatter patterns were recorded with the Oxford Instruments Aztec software package, and analysed within Oxford Instruments post-acquisition processing software Aztec Crystal (Version 2.1). Maps show quantitative EBSD data (Figs. 3e-

h, 4e-h). Grain maps depict individual grains (as previously defined) in random colours (Figs. 3e,f, 4e,f). Orientation maps show the crystal orientation at each analysis spot, colour coded according to the relative orientation of the crystallographic axes in relation to the sample *y*-direction (see colour key in Figs. 3h, 4h). A red colour denotes that the *c*-axis of the crystal is parallel to the sample's *y*-direction. As the colour moves away from red, as shown on the colour key, the orientation of that crystal has moved away from parallel. Grain orientations were also plotted on equal-area pole figures. These pole figures (Figs.

3i,j, 4i,j) are a 2D representation of a 3D feature; where a single point represents a line, here representing the specific axes of the individual grains. The orientation space on the pole figure mimics the spatial distribution across the shell. 3D diagrams of individual calcite unit cells show the orientation of individual crystals in space, and three crystals across each shell are highlighted on the maps and pole figures and presented as unit-cell diagrams, to show how the crystal orientation changes across the curvature of the shell (Figs. 3g,h, 4g,h). As represented by the 3D diagram unit cells, the *c*-axis is the long axis. The

crystallographic system of calcite (trigonal, hexagonal scalenohedral) dictates there is only one *c*-axis, but three *a*- and *m*-axes (Nicolas and Poirier, 1976) each perpendicular to the *c*-axis in each crystal, which is why there are more points on the *a*- and *m*-axis pole figures. Due to the tightly constrained orientation of many of the *c*-axes perpendicular to the growth surface, it is only possible to differentiate different grains by also taking the orientation data of the *a*- and *m*-axes into account, thus using the full crystallographic orientation data.






**Figure 2. Scanning electron micrographs of representative specimens from samples used in this study a,b,c) glassy _G. ruber_ from GLOW 22; d,e,f) frosty _G. ruber_ from IODP U1335A; g,h,i) glassy _M. crater_ from TDP 20; j,k,l) frosty _M. crater_ from DSDP 527. Scale bars for whole specimens (panels a,d,g,j) = 100 μm; scale bars in cross section micrographs (panels b,c,e,f,h,i,k,l) = 10 μm**



## 3 Results

### 3.1 General wall texture

Scanning electron microscope (SEM) images (Fig. 2) show some preliminary differences between glassy and frosty specimens of both species. Glassy specimens display a microgranular calcite texture in cross section, with sub-micron microgranules not
resolvable at this scale: a texture typical of good preservation. The texture differs in the frosty specimens, which possess blockier, larger (typically ~1 µm in diameter), more equant calcite structures in the test wall cross sections, that are more clearly visible in SEM. There are also even larger bladed crystals towards the outer edge of the frosty *M. crater* specimens. Forescatter electron (FSE) images show the different wall textures, and further indicate the degree of preservation in the two species (Figs. 3a-d, 4a-d). The *G. ruber* texture in the glassy specimen cross section shows the typical wall ultrastructure of a
cancellate species with topographic lows and highs in the test wall indicating pores and inter-pore ridges. The outer test wall and pore channels are less distinct in the frosty *G. ruber* specimen. The finger-like projections characteristic of the muricate wall texture are clearly visible in the glassy *M. crater* (Fig. 4a) and are present but less clearly defined in the frosty *M. crater* (Fig. 4c). In both glassy *G. ruber* and *M. crater*, bands that run parallel to the surface of the test wall are visible, representing growth layers, and these bands are exceptionally clear in the glassy *M. crater* specimen. This growth structure is not visible in
either frosty specimen. A major difference in the test structure of the two *M. crater* cross sections is the much thicker test wall in the frosty specimen (Fig. 4c) compared with its glassy counterpart (Fig. 4a). Darker sections of the images indicate areas with increased porosity (Figs. 3a-d, 4a-d), with both *G. ruber* and *M. crater* frosty specimens more porous than their glassy counterparts. In both frosty specimens, the areas of greatest porosity are towards the inside of the test wall, approximately one third of the way up from the inner surface, with less porosity observed on the outermost sides.

### 3.2 Grain geometry and microstructure

EBSD grain maps show foraminiferal calcite to be arranged in crystallographically aligned domains, or "mesocrystals". Grains are highlighted in randomly assigned colours, indicating the size and shape of these crystallographically aligned domains (Figs. 3e,f, 4e,f). These coloured microstructural maps are distinctly different between glassy and frosty specimens of both species. The glassy *G. ruber* has very well organised, elongate mesocrystals projecting perpendicular to the interior test wall. The
mesocrystals in the frosty *G. ruber* are less organised and less linear in shape, with more irregular shaped grain boundaries between domains. While there is little difference in the mean size of mesocrystals between frosty and glassy *G. ruber* (8.61 µm$^2$ and 9.92 µm$^2$ respectively), there is a notable difference in the size of the largest grains, with a maximum grain size in the frosty *G. ruber* specimen of 67.57 µm$^2$, compared with a maximum grain size of 110.48 µm$^2$ in the glassy *G. ruber* specimen.

The shape and size of mesocrystals in glassy *M. crater* is decidedly different to those seen in glassy *G. ruber*. Mesocrystals are much smaller, with a mean grain size of 1.94 µm$^2$ compared to the mean size of glassy *G. ruber* crystals of 9.92 µm$^2$. These domains do not appear to be organised as linear structures. The crystal geometry in the glassy and frosty *M. crater* specimens





is also distinctly different. The mean measured grain size in the frosty specimen is larger than in the glassy test (11.66 μm² and 1.94 μm², respectively), which is clearly visible in Figs. 4e,f, where the largest grains in the frosty *M. crater* specimen (231.08

μm²) are almost six times greater than the largest measured grain size in the glassy specimen (38.84 μm²). There is also much greater variation in the size of grains measured in the frosty *M. crater* compared with the glassy counterpart, with standard deviations in grain size area of 26.04 μm² and 3.63 μm² respectively. This difference in grain size correlates with the position within the test wall, from smaller grains at the inside edge of the shell to much larger crystals towards the outer edge of the shell. The geometry and shape of these larger grains on the outer edge of frosty *M. crater* differ strikingly from the smaller

grains. Large grains show 'fanning' grain boundaries, with grains becoming wider towards the outer rim of the shell. This is associated here with a lack of porosity (Figs. 4c,f,h).

Calcite twinning, defined as a 60° rotation around the *c*-axis, is present in both *G. ruber* and *M. crater* specimens, at a frequency much greater than would be expected by chance orientation (Appendix A). The glassy *G. ruber* exhibits twinning in 22% of

grains, whereas twinning is only displayed in 7% of grains in the glassy *M. crater*. Both frosty specimens show less extensive twinning, with 9% of *G. ruber* grains displaying twinning, and just 2% in the frosty *M. crater* (Appendix B).

### 3.3 Crystallographic orientation characteristics of mesocrystals

The mesocrystals in every specimen analysed (Figs. 3g,h, 4g,h) show a preferred crystallographic orientation of the *c*-axis (long axis) aligned perpendicular to the shell wall and the primary organic sheet (POS). The *c*-axis of crystals are most strongly

clustered on the pole figures of glassy *G. ruber*, with strong alignment also present in frosty *G. ruber,* but with greater spread in the points for the latter (Fig. 3i,j). In the *G. ruber* specimens in particular, the orientation of mesocrystals follows the geometry of the shell, as represented by the gradual change in colour on the IPF-Y coloured crystal maps (Figs. 3g,h). Individual unit cell schematics (red hexagonal prisms in Figs. 3g,h, 4g,h) show this grain orientation and how it changes with the curvature of the shell. Glassy *M. crater* also displays this general preferred *c*-axis orientation; however, the *c*-axis

orientation appears to differ where larger muricae are present (on either side of the specimen cross section), as seen in the greater spread of data on the pole figures and the wider range of colours on the IPF-Y plots (Fig. 4g), with *c*-axis alignment deviating by up to 70 degrees from perpendicular in these larger muricae. While the frosty *M. crater* has largely retained a similar crystallographic orientation to the glassy specimen with *c*-axes typically aligned perpendicular to the shell wall, more randomly oriented grains are interspersed, shown by the presence of different colours in the IPF-Y plot (Fig. 4h). The spread

of crystal orientation in *M. crater* specimens (Figs. 4i,j) is greater than that observed in *G. ruber* specimens. In all specimens, the orientations of *a*- and *m*-axes are more varied than *c*-axes.

### 3.4 Magnesium/Calcium Geochemistry

Electron microprobe analyser (EMPA) maps show clear, several micron-wide bands of alternating low and elevated Mg/Ca ratios in both glassy and frosty specimens of *G. ruber* (Figs. 3k,l). However, this banding is less well defined and generally



more homogenised in the frosty specimen of *G. ruber*. Absolute Mg/Ca ratios appear generally higher in the frosty specimen, with a clear band of high Mg/Ca ratios (up to 10 mmol/mol) towards the outside edge of the test (Fig. 3l). Alternating bands of higher and low Mg/Ca ratios are exceptionally clear in the glassy *M. crater* specimen, where chemical banding is prevalent throughout. The Mg/Ca ratios in the frosty specimen however are completely homogenised, consisting of low Mg/Ca calcite, with no distinct interspersed bands of higher magnesium concentrations.






**Figure 3. Forescatter images of 'glassy' (a,b) and 'frosty' (c,d) *G. ruber* specimens, with the area analysed highlighted by the orange box (b,d). Microstructural crystal maps (e,f) with grains highlighted in random colours to show grain size, shape, and position. IPF-Y coloured maps (g,h) showing the orientation of crystals with respect to the Y-direction of the sample. Pole plots (i,j) showing the crystallographic orientation of the mapped area. Selected grains highlighted show the change in orientation with the geometry of the shell, with unit cell schematics (g,h). EMPA-derived Mg/Ca maps are shown in (k,l).**





**Figure 4. Forescatter images of 'glassy' (a,b) and 'frosty' (c,d)** *M. crater* **specimens, with the area analysed highlighted by the orange box (b,d). Microstructural crystal maps (e,f) with grains highlighted in random colours to show grain size, shape, and position. IPF-Y coloured maps (g,h) showing the orientation of crystals with respect to the Y-direction of the sample. Pole plots (i,j) showing the crystallographic orientation of the mapped area. Selected grains highlighted show the change in orientation with the geometry of the shell, with unit cell schematics (g,h). EMPA-derived Mg/Ca maps are shown in (k,l), previously published in John et al. (2023).**



## 4 Discussion

### 4.1 Microstructure and geochemistry of primary biogenic calcite


Glassy specimens are generally assumed to have been subject to negligible post-mortem modification (Sexton et al., 2006). As such, the microstructure of glassy specimens is taken as representative of the primary biogenic calcite microstructure for each species. Despite having different wall textures, the incremental addition of calcite layers, as evidenced by the internal
banding shown in FSE images (Figs. 3a, 4a), is present in both *G. ruber* and *M. crater.* The fact that these intricate cross-sectional details are still visible, confirms that these specimens have experienced little diagenetic alteration. We observe a mesocrystal structure in the well-preserved, glassy specimens of both species, although there are differences in the mesocrystal geometry of each wall texture. The predominant orientation of crystal *c*-axis oriented perpendicular to the test chamber surface for both *G. ruber* and *M. crater* is consistent with existing studies of foraminiferal calcite crystal orientation (Pabich, Vollmen
and Gussone, 2020; Arns et al., 2022; Lastam et al., 2023). The mesocrystals we observe are made up of microcrystallites, all oriented within 10º of each other. These crystallites are formed as layers of biogenic calcite (the previously noted internal banding), which are deposited over the whole test each time a foraminifer precipitates a new chamber (Fehrenbacher et al. 2017). These depositions propagate outwards from the POS, creating large elongate mesocrystals in *G. ruber* (Fig. 3e), and smaller, less elongate mesocrystals in *M. crater*, which may be a function of the spine like projections of the muricate wall
texture throughout the test wall (Fig. 4e) (Pearson et al. 2022). The mesocrystals of glassy *G. ruber* continue through the compositional banding (Fig 3), suggesting that each successive layer of calcite, retains or templates the same orientation as the crystals laid down before it. Lastam et al. (2023[a]) found that the inclusion of organic layers throughout the test (the darker layers highlighted by our FSE images) does not impede the addition of consistently oriented calcite growths after each chamber formation, and may even contribute to maintaining the consistent direction of growth by templating the orientation laid down
before each additional chamber.

There is an apparent increase in twin boundary prevalence in the modern *G. ruber* species compared with the Paleogene *M. crater*. Previous work has identified calcite twinning as a strongly developed feature in three other modern perforate planktonic foraminifera species (*G. sacculifer, O. universa*, and *P. obliquiloculata*), whereas twinned calcite although present, is not a
predominant structural characteristic in the two microperforate species studied (Lastam et al, 2023[b]). The apparent increase in twin prevalence we observe between the extinct Paleogene species *M. crater* and the extant species *G. ruber* may be reflective of an evolutionary biological development. Twinned crystal boundaries are structurally stronger (Lu, 2016) and it is possible that perforate planktonic foraminifera have developed an affinity for twinned crystal boundaries to develop a stronger biogenic structure. To test this hypothesis would require further investigation widening the data set to a significantly higher number of
specimens and species.



We observe alternating bands of high and low Mg/Ca ratios in both well preserved *G. ruber* and *M. crater* specimens at high spatial resolution, as has been described in other studies (e.g. Sadekov, Eggins and De Deckker, 2005; Fehrenbacher et al., 2017; John et al., 2023). The internal banding of differing calcite chemical composition is thought to be a result of biological

processes, including chamber formation (Jonkers et al., 2016), presence of organic membranes (e.g. Kunioka et al., 2006; Branson et al., 2016; Bonnin et al., 2019), and biomineralisation on a diurnal cycle after the deposition of the final chamber, during test thickening (e.g. Spero et al., 2015; Fehrenbacher et al., 2017). This lateral geochemical fabric gives credence to the lateral templating observed by consistent crystal orientation of subsequently deposited layers of calcite. The high resolution of alternating higher and low Mg/Ca banding observed also implies we are observing well preserved specimens.

**4.2 Impact of post-depositional modification on primary microstructure and geochemistry: signatures and processes**

Assuming the glassy specimens are the most representative of original unaltered foraminiferal calcite, comparing the microstructure of these specimens alongside frosty samples allows us to make inferences about post-mortem changes. In the following, we identify three main processes, namely (i) dissolution, (ii) interface coupled fluid mediated replacement reactions of dissolution and re-precipitation, and (iii) overgrowth; which we argue to be responsible for the observed range of changes

both in geochemistry and microstructure. A summary of these processes and their different effects on foraminiferal calcite are presented in Table 2 and schematically in Figure 5.

Frosty specimens of both species display increased porosity throughout the test wall (Figs. 3c, 4c) that is not present in either glassy specimen, suggesting post-depositional modification has occurred via dissolution of test calcite (Pearson et al., 2007;

Pearson and Burgess, 2008; Johnstone et al., 2011). This dissolution is most observable on the innermost side of the test wall, permeating up to a third through the thickness of the test, starting roughly in line with the site of the POS. Dissolution occurs in corrosive waters under-saturated with respect to carbonate, resulting in a loss of test material (Fig. 5b, Table 2). Previous work has suggested that the POS and other organic linings provide an opportunity for fluid flow within the test wall (Cisneros-Lazaro et al., 2022), and the concentration of dissolution effects near the site of the POS could indicate that corrosive water

may indeed have exploited this susceptible area in the test walls of our frosty specimens. This is a significant observation, because calcite associated with the POS may be enriched in magnesium (Branson et al., 2016). Therefore, preferential dissolution of this region may contribute to the relationship between planktonic foraminiferal shell weight and mean test Mg/Ca (e.g., Rosenthal and Lohmann, 2002).

In addition to dissolution, our frosty *G. ruber* specimen displays evidence of interface coupled fluid mediated replacement reactions (Putnis, 2009) (Fig. 5c, Table 2); commonly referred to as "recrystallisation". In this work, we use the term interface coupled fluid mediated replacement reactions, consisting of paired dissolution and re-precipitation to better unpack the specific processes associated with post-depositional modification of test calcite, instead of "recrystallization", which has previously been applied to encompass multiple forms of post-depositional change to foraminifera tests. In our frosty samples, increased



porosity is combined with a limited change in grain shape, diagnostic features of paired dissolution-precipitation events (Sexton et al., 2006). Such fluid mediated replacement reactions occur when the local fluid or surrounding waters (either within the test, as intra-test fluid films interacting with the test calcite, or pore waters surrounding foraminifera tests) are under-saturated with respect to carbonate; resulting in dissolution until a local oversaturation of the surrounding fluid film with carbonate ions, gained from the parent material (Fig. 5c). These surrounding carbonate ions are then re-incorporated into the

test forming new material, as it re-precipitates at the fluid-solid interface (Putnis, 2009). This sequence then repeats if calcite porosity allows local fluid exchange. Newly re-precipitated calcite remains porous, enabling this continued exchange. The non-linear shape of mesocrystals in both frosty specimens, suggests that fluid-mediated replacement reactions are occurring at intra-test grain boundaries, facilitated by extremely thin films of fluid between grain boundaries, and is not occurring solely as a linear process from the outside of the test to within. Our results show that although re-precipitation is an inorganic post-

depositional process, it is somewhat "templated" by the original biogenic structure, resulting in irregular grain boundaries and a change in crystal shape, but a retention of the general crystallographic orientation with relatively consistent orientation plotting of *c*-axes on the pole figures between glassy and frosty specimens, as observed in our frosty *G. ruber*. This observed templating of the original biogenic structure is a typical feature of products of coupled dissolution-precipitation reactions (Spruzeniece, Piazolo and Maynard-Casely, 2017). This observation may shed some light on the mechanisms which alter

foraminiferal test calcite, with implications for our understanding of the preservation of test geochemistry of frosty foraminifera.

With interface coupled fluid mediated replacement reactions, there is a partial loss of original geochemical signals as the original test calcite is replaced by material with a chemistry governed by a mix of local fluid chemistry and original shell

chemistry, which is reflected in the decrease in resolution of alternating bands of high and low Mg/Ca ratios compared with the glassy specimen of *G. ruber*. The retention of some Mg/Ca banding however, reinforces the notion that Mg/Ca banding can be preserved in recrystallized foraminifera tests, to a degree which may remain useful to geochemical proxy generation (Staudigel et al., 2022). The higher Mg/Ca ratios observed in our frosty *G. ruber* compared with our glassy *G. ruber* are uncharacteristic of post-depositionally modified foraminiferal calcite (Regenberg et al., 2014). We note that both of our frosty

foraminifera display a band of higher Mg/Ca (~ 8-10 mmol/mol) towards the outer edge of the test wall, which potentially indicates the presence of an authigenic coating (Barker, Greaves and Elderfield, 2003). Our samples were not chemically cleaned to remove coatings prior to analysis, as to avoid loss of carbonate material and to avoid altering the structure of the test; so we cannot comment on the implication of this for palaeoclimate reconstructions.

A simple model has been used to illustrate one possible reason why planktonic foraminiferal mean Mg/Ca appears to be more robust to post-depositional modification than $\delta^{18}O$ (Staudigel et al., 2022). In this model, the composition of the newly precipitated calcite depends on the temperature-dependent magnesium partition coefficient and the fluid chemistry (i.e. Mg/Ca). However, because the fluid is present as extremely thin films of fluid moving through the foraminiferal test, their



composition soon evolves from the original pore water composition, with this evolution a function of the reaction and diffusion
rates. In the case of Mg/Ca, the system soon becomes "rock-buffered" which limits the potential for major changes in intra-
test mean Mg/Ca, at least for early-stage alteration (Staudigel et al., 2022). This model proposes that during early-stage
alteration, magnesium cations are primarily rearranged within the foraminiferal calcite lattice, smearing out the well-defined
banding, but this is not associated with a major change in whole test Mg/Ca, which is critical for the Mg/Ca paleothermometer.
This geochemical preservation is in contrast to planktonic foraminiferal δ$^{18}$O, which has much greater opportunity to be reset
by the water films passing through the test. Our observations of the frosty *G. ruber* are consistent with model predictions for
early stage alteration (Staudigel et al., 2022). The diagenetic model suggests that ongoing modification processes will
eventually destroy the banding, and the Mg/Ca of the whole test will increase as the calcite reaches equilibrium with the burial
temperature and porewater chemistry. The lack of magnesium banding but generally low Mg/Ca of the frosty *M. crater*
specimen indicates that it is further along this proposed diagenetic pathway. The extent to which the mean Mg/Ca of this
individual test has been shifted is currently impossible to determine, so a cautious approach would be to avoid using such a
specimen for palaeoclimate reconstructions.

The frosty *M. crater* specimen shows evidence of yet another post-depositional modification process; overgrowth formation
(Fig. 5d, Table 2) (Pearson et al., 2007; Pearson and Burgess, 2008; Kozdon et al., 2011). A striking difference between glassy
and frosty *M. crater* specimens is the increased test thickness. This increase in thickness is associated with a change in
microstructure, i.e. fanning of grains (Figs. 4f,h, 5d) and a loss of porosity in the same areas, which could in part be due to the
ontogenetic build-up of calcite around the muricae, as is typical of species with this wall texture (Pearson et al., 2022).
However, this thickness is not consistent across the entire surface of the test, with additional calcite precipitation present on
the corners of chambers especially, and some chamber walls to a greater extent than others (Fig. 4d, yellow arrows), which is
not typically seen on well-preserved specimens. The microgranular structure typical of primary biogenic calcite, as observed
in the glassy *M. crater* specimen, is absent in the frosty *M. crater*, which has a crystal geometry composed mostly of coarse
crystals with straight grain boundaries, and evidence of crystal fanning. This is not consistent throughout the entire thickness
of the shell wall however, with the innermost side still possessing smaller calcite grains. The horizon where the grain sizes and
shapes change from smaller, more random grains, to larger, more linear grains, appears to be at the site where the greatest
dissolution has occurred (Fig. 4c,f, white stippled line). Based on the lack of porosity, the infill of previously present porosity
and voids, and the fanning nature of the grains, we suggest this test thickening is the result of post-depositional overgrowth by
inorganic calcite.

Such overgrowth occurs when foraminifera are surrounded by fluid that is oversaturated with respect to carbonate, and new
crystal growth occurs using the remaining shell structure as nucleation sites. Inorganic overgrowth is typically seen in
experiments with clear late stage oversaturation (e.g. Perdikouri et al., 2013). The fanning nature observed of the coarse grains
is typical for so-called growth competition during crystal growth where several grains compete, and is known from inorganic



crystal growth into an oversaturated fluid (e.g. Bons et al. 2012). The grain size coarsening we observe towards the outer edge of the test wall (Figs 4c, 5d) is also indicative of extensive "recrystallization" (Pabich, Vollmen and Gussone, 2020) or, more

precisely, post-depositional modification by overgrowth. Some of the largest grains we observe on the outer edge of the test wall are likely to be representative of individual large crystals, as opposed to mesocrystals of well-aligned microgranules as observed in the glassy specimen. The smaller grains we observe nearer to the *M. crater* internal test wall may not have been subject to the same level of post-depositional overprinting. Lastam et al. (2022[b]) observe different calcite microstructure on either side of the POS in tests of *Pulleniatina obliquiloculata*, with blocky crystals on the inner-side, and fractal/dendritic

mesocrystals from the POS to the outer side. While clear differences in test crystallographic structure are observed in our EBSD maps of frosty *M. crater* (also visible in cross-sectional SEM (Fig 2l)), our data for well-preserved *M. crater* do not suggest the presence of different crystal structures within this wall type. Such a difference if present could explain the significant difference in crystal size from the innermost side to the outer edge of the test in our altered *M. crater* specimen, however we attribute this feature to be most likely due to differential effects of post-depositional modification across the test

wall, with dissolution and re-precipitation at the inner-side, and overgrowth characterising the outer edge of the test. The general crystallographic orientation of the overgrowth observed in frosty *M. crater* indicates a retention of the original grain orientation (Figs. 4h,j) consistent with nucleation of inorganic calcite on the original grains, hence retaining the original preferred orientation. This type of growth is referred to in the crystallographic literature of vein growth (i.e. crystal growth into free space in fluid filled rock cracks) as syntaxial growth (Bons, Elburg and Gomez-Rivas, 2012).


Alternating high and low Mg/Ca bands are absent in the frosty *M. crater* specimen, with test chemistry now highly influenced by the chemistry of surrounding waters (Fig 5d,e). The loss of high Mg/Ca ratio bands during dissolution lowers the bulk Mg/Ca ratio of the test, a common observation in foraminiferal calcite that has undergone dissolution (Regenberg et al., 2014). The Mg/Ca composition of the calcite overgrowth observed in the frosty *M. crater* appears to be the same composition as the

underlying re-precipitated material. This is an unexpected observation, as the inorganic overgrowth was unlikely to form at the exact same time as the re-precipitated material below, and raises the question as to whether calcite composition, potentially as a function of calcite microstructure, could be templated in overgrowth formation. The presence of magnesium in calcite is known to influence microstructure (Folk, 1974). High Mg in calcite facilitates vertical alignment of crystals, with "Mg-poisoning" (the addition of $Mg^{2+}$ ions to the side of a growing crystal, causing surrounding $CO_3^{2-}$ sheets to scrunch together to

accommodate such $Mg^{2+}$ ions) inhibiting sideward growth (Folk, 1974). This is supported in our data, most especially with the strict vertical alignment of mesocrystals in our glassy *G. ruber* specimen. Contrarily, low Mg calcite, enables rapid sideward growth (Folk, 1974). The preferential removal of higher Mg calcite during dissolution may therefore facilitate the production of less linear, wider, and more equant crystals, as observed in our frosty *M. crater* specimen, with Mg poisoning no longer limiting calcite growth during re-precipitation and overgrowth formation. The difference in chemical heterogeneity

preservation between *G. ruber* and *M. crater* reflects how the frosty specimens of these species have been subject to differing degrees and processes of post-depositional modification. While the Mg/Ca proxy is thought to be more robust to the effects of





diagenesis than the $\delta^{18}O$ proxy (Kozdon et al., 2013; Staudigel et al., 2022), our EBSD data supports the idea that pervasive microstructural alteration may ultimately result in the loss of Mg banding and the alteration of original bulk Mg/Ca values during late-stage post-depositional modification.


The processes described here; (i) dissolution, (ii) fluid mediated replacement reactions of paired dissolution-precipitation, and (iii) overgrowth (Table 2, Fig 5), may occur in combination, as observed in the frosty *M. crater* specimen. The terms "recrystallization" and "diagenetic alteration" frequently used in foraminiferal research are descriptive terms, that are commonly used to encompass these specific processes which, as demonstrated, can modify test microstructure and

geochemistry to differing degrees.







**Figure 5. Schematic diagram illustrating different post-depositional processes affecting foraminifera test chemistry and microstructure, a) original test; b) dissolution; c) replacement: coupled dissolution and precipitation; d) late stage replacement; e) overgrowth**



**4.3 The importance of post-depositional history for modification of original test textures and chemistry**

Glassy specimens from sites known to host well preserved planktonic foraminifera have been targeted to minimise the potential effects of diagenetic alteration (Nicholas et al., 2006; Bown et al., 2008; Birch et al., 2013). The sites selected for providing glassy specimens were both above the lysocline (palaeo-water depths < 2000 m) (Kroon et al., 2009; Nicholas et al., 2006) with bottom waters oversaturated with respect to pure calcite ($\Delta[CO_3^{2-}]>0$ µmol kg$^{-1}$,$\Omega>1$) (Regenberg et al., 2014), preventing severe dissolution of foraminifera tests. Additionally, these sites consist predominantly of impermeable clay rich sediments,

which rapidly encase foraminifera tests upon deposition, protecting them from future pore-water interactions. Our post-depositional model (Fig. 6) highlights these key factors contributing to excellent preservation potential, namely, waters supersaturated with respect to calcite and clay rich sediments (Berger, 1970; Berger, 1979; Schiebel et al., 2007; Bown et al., 2008; Pearson and Burgess, 2008). Following our analyses, both glassy specimens from sites that meet these criteria do indeed show little to no evidence of diagenetic alteration, either microstructurally or geochemically. These sites offer preferential

criteria for geochemical proxy generation, and for studies on the original biogenic growth of foraminifera.

By contrast, the sample materials that were selected for collection of our frosty specimens were deposited below the lysocline (palaeo-water depths ~ 4000 m) with corrosive bottom waters presumably undersaturated with respect to CaCO$_3$. They were also deposited in open ocean settings, with little terrigenous clay input, and the predominant sediment encasing these tests after

deposition was pelagic carbonate ooze, a relatively permeable sediment type offering little protection from corrosive pore water infiltration. Despite deposition in similar oceanographic settings, below the lysocline and within carbonate ooze, the frosty *M. crater* appears to have undergone more extensive post-depositional modification than the frosty *G. ruber*, as described above. Our frosty *G. ruber* specimen shows evidence of interface-coupled fluid mediated replacement reactions of paired dissolution and re-precipitation, while the frosty *M. crater* specimen displays evidence of dissolution, interface-coupled

fluid mediated replacement, and additional calcite overgrowth. These processes occur during interaction of the foraminifera shells with corrosive bottom waters of the seafloor, and corrosive pore waters in the sediment stack. While this interaction occurred for both frosty *G. ruber* and *M. crater*, *M. crater* likely experienced a greater degree of pore water interaction than *G. ruber* due to the longer period the shell has been within the porous sediment stack, more than 30 million years longer than *G. ruber*. Longer time in this calcareous sediment likely contributed to the formation of the calcite overgrowth by increased

exposure to mobile carbonate ions and associated chemical oversaturation due to the high concentration of carbonate in the sediment. While time is a contributing factor to the preservation quality of these samples, the preservation of our glassy *M. crater* individual, of a similar age to the frosty *M. crater* specimen, suggests that porewater chemistry and sediment composition must be at least, if not more important contributing factors.

It is also possible that the muricae present on the wall surface of *M. crater* specimens provided structural focal points for calcite overgrowth (Sexton et al. 2006; Kozdon et al., 2011). In our EBSD maps we do not see the same overgrowth pattern in our



frosty *G. ruber* specimen; as such our data suggest that species' wall texture may be a contributing factor as to how foraminifera are affected by different diagenetic processes. It has also been suggested previously that some portions of a test, for example the base of muricae which have low porosity, may be protected from major post-depositional modification and geochemical alteration (Kozdon et al., 2011), supporting this hypothesis. Our depositional model (Fig. 6) captures the key factors contributing to degraded preservation potential, namely, waters under saturated with respect to pure calcite and the absence of clay rich sediments. Following our analyses, both frosty specimens from sites which were deposited in pelagic carbonate oozes below the lysocline show extensive evidence of post-depositional modification, both microstructurally and geochemically. We recommend caution, particularly when interpreting microstructural details and environmental proxy information, when working on materials from such sites.

## 5 Summary

Electron backscatter diffraction analysis (EBSD) enables the visualisation and quantification of foraminiferal test microstructure at the crystallographic scale, providing insight into biomineralization processes as well as specific processes of post-depositional modification. Our EBSD mapping highlights how the processes of dissolution, interface-coupled fluid mediated replacement reactions of dissolution and re-precipitation, and inorganic overgrowth can differentially affect test microstructure. Pairing these microstructural data with EMPA maps of Mg/Ca concentrations, correlates impacts on test microstructure with changes in test geochemistry. In glassy specimens, we observe a mesocrystal structure, of crystallographically aligned domains made up of smaller units of microgranular calcite, all similarly aligned, with a general preferred orientation of mesocrystal *c*-axes aligned perpendicular to the test wall. Our data shows when post-depositional inorganic calcite forms either by fluid-mediated replacement reactions or by overgrowth, this orientation of original biogenic calcite is generally maintained with *c*-axes aligned perpendicular to the test surface, producing a pervasive new crystalline structure, despite some change in size and shape of mesocrystals. Our frosty *G. ruber* specimen contains relic magnesium banding that cuts across the mosaic of granular mesocrystals, which supports the suggestion that fluid-mediated replacement reactions may be relatively "rock-buffered", or contained, thus protecting some geochemical proxies such as Mg/Ca palaeothermometry. However, our frosty specimens display different degrees of post-depositional modification, both microstructurally and geochemically, which underscores the importance of a detailed preservation assessment before sites are targeted for paleoclimate reconstructions. Our frosty *M. crater* displays large bladed calcite overgrowths growing from the muricae. These large inorganic crystals are not seen in our frosty *G. ruber* specimen, implying that in addition to the depositional environment, the structure of the primary test may also influence diagenetic susceptibility. Providing novel insights into the extent of test modification, EBSD can be used to unpack post-depositional modification processes, and inform the utility of individual microfossil samples in geochemical research.





**Figure 6. Post-depositional model, representing the depositional environments of each of the specimens studied**




## Appendix A – Frequency of twin boundaries

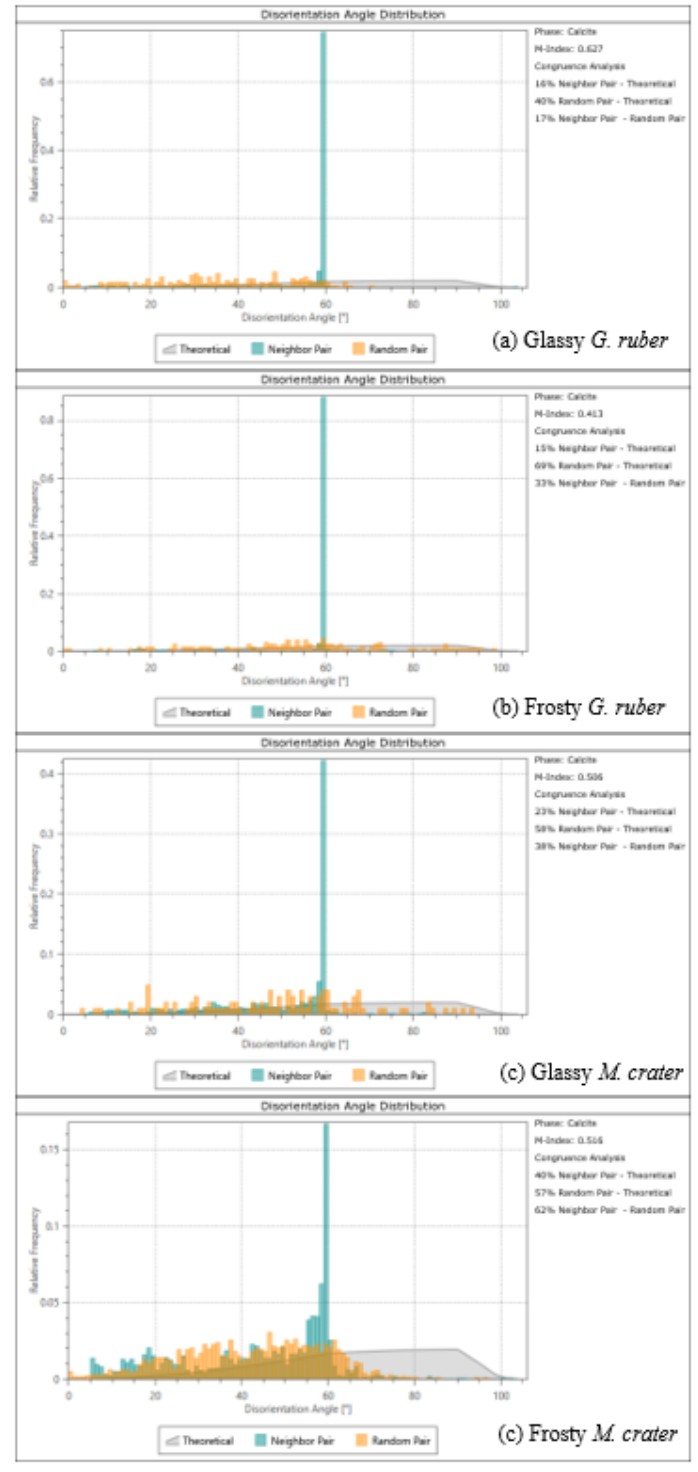

Disorientation angle distribution charts showing the 'Neighbor Pair Distribution': plotting the misorientation data between neighboring points in the map, alongside the 'Random Pair Distribution': plotting misorientation data between randomly chosen points in the data set, and the 'Theoretical (Mackenzie-Plot)': showing the theoretical distribution that would be expected from a random set of orientations. This shows that twin boundaries (neighboring pairs of points with a misorientation of ~ 60 degrees) occur at a significantly higher frequency than would be expected by theoretical chance occurrence.




## Appendix B – Abundance of twin boundaries

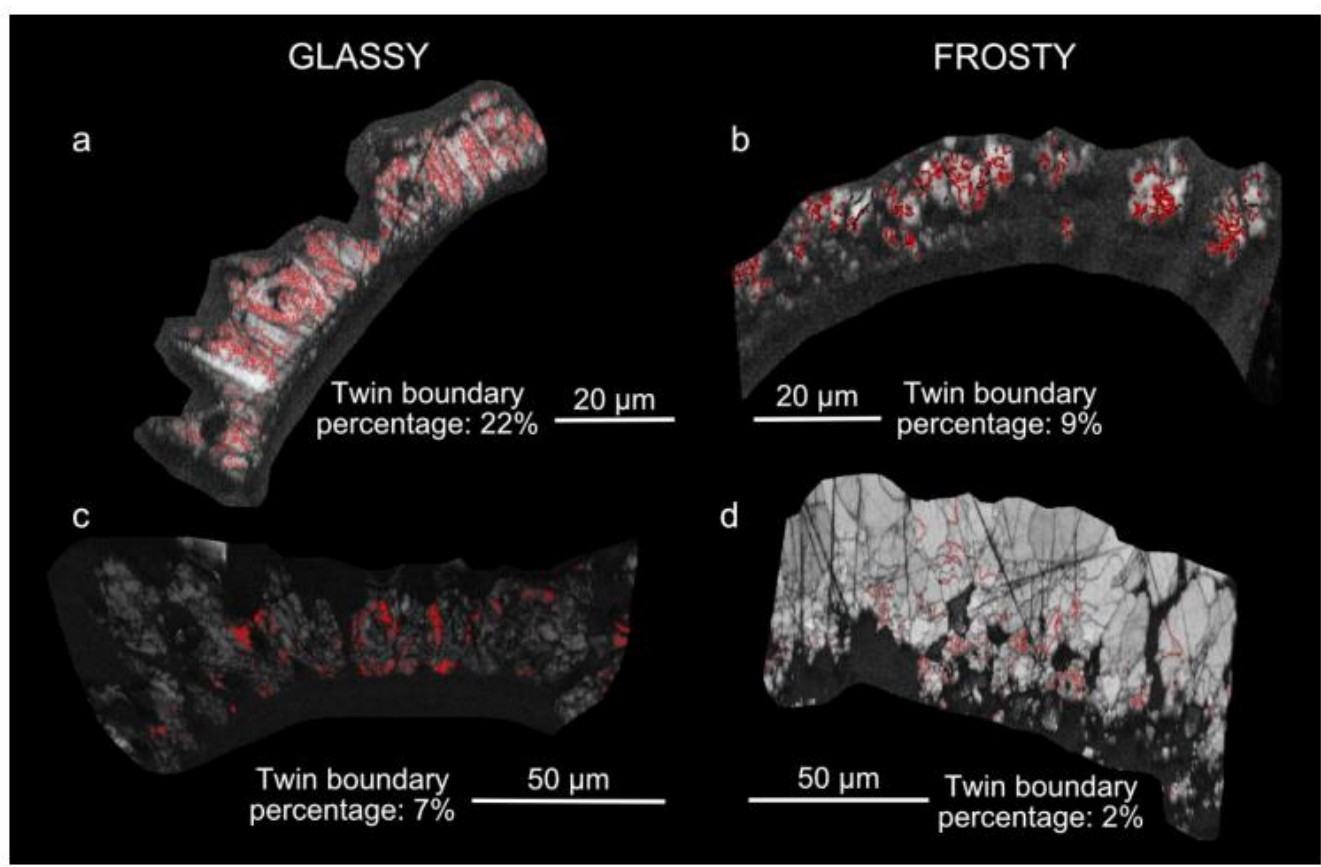

Analysis results showing abundance of twinning in the four specimens studied; band contrast (pattern quality) maps with twin boundaries (60° rotation around *c*-axis) shown in red. a) glassy *G. ruber*, 22%; b) frosty *G. ruber*, 9%; c) glassy *M. crater*, 510   7%; d) frosty *M. crater*, 2%. Numbers refer to the proportion of calcite grains that have twin boundaries.



**Author Contributions**

Conceptualization: FAP, SP, TA

Methodology: FAP, SP, EHJ, RW

Investigation: FAP, SP, EHJ, RW

Visualization: FAP, EHJ

Supervision: FAP, SP, TA

Writing—original draft: FAP

Writing—review & editing: TA, SP, EJ, PNP, CHL, RW

**Competing Interests**

The authors declare that they have no conflict of interest.

**Acknowledgements**

This work was supported by the Natural Environmental Research Council (Studentship grant NE/S007458/1), and the Natural Environment Research Council grant NE/P019102/1 to C.H. Lear. The research uses samples supplied by the

International Ocean Discovery Program. Acquisition of glassy foraminifera from the GLOW cruise was funded through NERC grant NE/F523293/1 to P.N.P. and from TDP 20 through NERC grant NE/B503225/1 to P.N.P. We thank the Tanzania Commission for Science and Technology and Tanzania Petroleum Development Corporation for their assistance in this work, Harri Wyn Williams (University of Leeds, UK) and Ben Buse (University of Bristol, UK) for their invaluable technical support.




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
