# Peer review of "Electron backscatter diffraction analysis unveils foraminiferal calcite microstructure and processes of diagenetic alteration"

_EGUsphere, 2023_

## Author Response (AR2)

**Author's Response to the Review Comments and the Editor**

**We have completed all of the changes listed below during revision, and have updated the manuscript with the requested technical corrections:**

**- We have added references for species time ranges**

**- The spelling of nannofossil has been corrected**

**- Line 116 - beginning of the sentence, EBSD has been spelled out**

**1) Reviewer Comments from Chiara Consolaro, and our responses highlighted in bold:**

I reviewed the paper Egusphere-2023-2213: Electron backscatter diffraction analysis unveils foraminiferal calcite microstructure and processes of diagenetic alteration by A. Procter et al., and I think that it is a very interesting and important topic, very well presented and well written.

I have only some minor comments to the manuscript before it can be accepted.

**We thank the reviewer for their kind words and positive feedback on our manuscript.**

Line 15: "data shows" it is correct but maybe it would be preferable to use it in its plural form, so **data show.**

**This has now been changed to 'data show'.**

Line 29: instead of "planktic" foraminifera is better to use **planktonic** like it has been done in the rest of the paper.

**We have replaced 'planktic' with 'planktonic'.**

Line 63: maybe substitute ";" with ", "after the quotes in brackets.

**We have substituted the semi-colon for a comma here.**

Lines 205-206: is there a reason why 'fanning' grains are associated with a lack of porosity or is it just an observation? Maybe elaborate further?

**We have clarified that this was an observation in our results: "These fanning grains are observed here to be associated with a lack of porosity (Figs. 4c,f,h)." (Now on line 215)**

Lines 215-216: rewrite or clarify the sentence "with greater spread in the points for the latter".

**We have re-written this sentence to read: "The *c*-axis of crystals are most strongly clustered on the pole figures of glassy *G. ruber*, with strong alignment also present in frosty *G. ruber*, despite a greater spread in the orientation data of frosty *G. ruber* (Fig. 3i,j)." (Now on lines 223-226)**

Line 217: state what does the abbreviation IPF-Y stands for when you mention it for the first time.

**We have added the explanation of the IPF-Y abbreviation into the methods section 2.4, where we describe these IPF-Y orientation maps. Lines 152-155 now read:**

**"Inverse pole figure colour maps showing crystallographic orientation in the Y direction (IPF-Y maps) show the crystal orientation at each analysis spot, colour coded according to the relative orientation of the crystallographic axes in relation to the sample *y*-direction (see colour key in Figs. 3h, 4h)."**

Figure 3 and 4: (g, h) the axes and the letters in the 3D diagram unit cells are not well visible, please make them more visible. (i,j) The plots of the selected grains on the pole plots represented in orange and yellow are difficult to see, maybe change color or make the symbols bigger?

**We have edited the 3D diagrams to make the labels clearer. We have also increased the size of the selected grains on the pole plots and changed the colour.**

Line 359: please explain better what do you mean with "further along this proposed diagenetic pathway".

**We have changed this sentence to read: "The lack of magnesium banding and generally low Mg/Ca of the frosty *M. crater* specimen indicates that it has experienced a greater degree of diagenetic alteration." (Now on lines 367-368)**

362: maybe substitute ";" with ":" after process, or reformulate the last part of the sentence.

**The semi colon has been replaced with a colon.**

Lines 409-414: maybe make a reference and clear connection with the last part of Fig. 5 (d and e)?

**We have added in references to Figs. 5d and 5e now within text lines 419 and 423, and also referenced earlier figures (Figs 3e,g, and 4f,h,) which highlight this point well.**

Appendix A- Frequency of twin boundaries: it is not possible to read the text on the graphs, maybe increase the quality of the figure?

**The size and quality of the figure has been increased to make sure the text on the figures is readable.**

Line 698: add a space after the reference of Putnis A., 2009

**We have added a space after this reference.**

**2) Reviewer Comments from Adriane Lam, and our responses highlighted in bold:**

The manuscript by Proctor et al. uses electron backscatter diffraction to investigate post-depositional processes that affect foraminiferal tests. This manuscript is extremely well-organized, well-structured, and scientifically important to everyone who use foraminiferal calcite in geochemical analyses. Below are a few comments:

**We thank the reviewer for their kind words and enthusiasm for the work we have presented.**

Throughout, either be consistent with use of 'planktic' or 'planktonic'

**Planktonic is now used exclusively throughout the manuscript.**

Line 92: Spell out 'Globigerinoides' at the beginning of the sentence

**This sentence now opens "*Globigerinoides ruber*".**

Line 92-93: the pforams@mikrotax site lists ruber as evolving within 10.46-11.63 Ma (Chaisson and Pearson, 1997)

**We have revised the date to 15.12 Ma to be consistent with Lamyman et al. (in prep), *Phylogeny of Planktonic Foraminifera*; however we note that the origin of *G. ruber* will be subject to revision as part of the Neogene and Quaternary planktonic foraminifera working group (https://www.mikrotax.org/pforams/pf-pages/NeogeneWorkingGroup.php)**

**Lamyman et al. use Stanley, Wetmore and Kennett (1988) for the dates for *G. ruber*.**

**'Stanley, K. Wetmore, L. and Kennett, J. P., 1988. Macroevolutionary differences between two major clades of Neogene planktonic foraminifera. *Paleobiology*,14, 235-249.'**

In the Methods section, it would be helpful to have a short paragraph about the lithology at each site, specifically within the interval from which specimens were taken for analyses.

**The lithology of core material from each site has been added to Section 2.1 Sample material and preparation.**

**After the line: "Foraminifera were recovered from core material known to contain foraminifera specimens with differing degrees of preservation (Fig. 1; Table 1)." On line 95, we have added in the following:**

**The sediments from both Tanzanian cores, TDP 20, and GLOW 22 are rich in clay (Kroon et al., 2009), with the principle lithology of our section from TDP 20 comprising soft light olive grey clays with mottled yellowish orange sandy clay (Nicholas et al., 2006). Our core samples from site U1335A and DSDP 527 are contrarily characterised by nanofossil ooze, with ~ 60% CaCO3 wt% in our section from U1335A (Pälike et al., 2010), and our DSDP section characterised by**

**nanofossil ooze and nanofossil chalk, has a calcium carbonate content fluctuating around 90% (Moore et al., 1984).**

Table 1: Would be helpful to also include a column with the water depth

**Water depths have been added to Table 1.**

Figures 5 and 6 are awesome!

**We thank the reviewer for their very kind praise of Figures 5 and 6.**

**3) Editor comments from Chiara Borrelli, and our responses in bold**

Public justification (visible to the public if the article is accepted and published):

This manuscript presents some very interesting data regarding the diagenetic alteration of two different species of planktonic foraminifera, Globigerinoides ruber and Morozovella crater. These species are representative of different geological ages, as well as shell (test) wall type. The work has been reviewed by two experts. They both agreed that the manuscript is well-written, the results are interesting, and the manuscript deserves to be published in BG after minor revisions. I concur with the reviewers' assessment. The manuscript is definitely well-written and easy to follow. The results are interesting and appropriate for publication in BG. However, I do have some concerns regarding the rather limited number of samples analyzed (two specimens/species; one glassy, one frosty). I appreciate that the analyses performed are quite detailed and time-consuming (although the EMPA Mg/Ca maps for M. crater come from a different study) and that it might not be feasible to analyze more specimens. However, I would like to see some additional discussion about how representative the dataset presented is in the context of the species analyzed and how applicable the findings are to other planktonic foraminiferal species. In addition, I have some minor comments as detailed below. I invite the authors to consider all the feedbacks received, to improve the discussion of the representativeness of their results, and to resubmit a revised version of their work after what I believe to be minor to moderate revisions.

**We thank the editor for their assessment of the manuscript, and for agreeing to publish this work in Biogeosciences after minor revisions.**

**From line 434, we have added in a short discussion about how representative our dataset is, highlighting that our data may not be fully representative of the species we selected due to the small sample size. We highlight that the exceptional preservation of our glassy specimens provides a good basis for our assumption that these specimens are good representatives of each species, however also highlight the need for future studies to analyze replicates, and also more species.**

**This was unfortunately beyond the scope of this study. Throughout the manuscript we also highlight that future work should seek to expand the dataset. E.g. lines 288, 443, 493**

Minor comments:

Line 80: SEM should be spelled at line 80 (first time the acronym is used) rather than at line 99.

**SEM has now been spelled in its first use on line 80**

Line 98: please, provide a brief explanation of why methanol was used to clean M. crater specimens.

**This was due to M. crater being cleaned in a different lab. We have clarified in the text: "methanol for *M. crater* specimens, dependant on lab protocol"**

Line 101: 'foraminifera specimens' should be 'foraminiferal specimens'. This comment applies to the entire manuscript.

**Both occurrences of 'foraminifera specimens' have been changed to 'foraminiferal specimens'**

Lines 103-106: for the silicon carbide disks, diamond paste, and colloidal silica, please provide the name of the manufacturers.

**The name of the manufacturers has been added within the text**

Section 2.3 – quantitative Mg and Ca maps were done by EMPA. Did you calibrate the instrument using a standard? If you did, please specify which one. If you did not, please provide some additional details about how quantitative maps were generated.

**We have added detail that we used in-house standards to calibrate the instrument on Line 136: "Calibrations for quantification were carried out for Ca on an in-house calcite mineral standard and for Mg on an in-house dolomite standard."**

Line 144: is 'quantitative EBSD data' the best way to describe these data?

**This has been changed to 'EBSD data'.**

Line 186: Because it is the beginning of the sentence, please spell out EBSD.

**EBSD has been replaced with Electron backscatter diffraction here**

Table 2 is missing.

**Table 2 has been added into the body of the manuscript**